# Integrating Experimental and Computational Analyses for Mechanical Characterization of Titanium Carbide/Aluminum Metal Matrix Composites

**DOI:** 10.3390/ma17092093

**Published:** 2024-04-29

**Authors:** Waqas Farid, Hailin Li, Zhengyu Wang, Huijie Cui, Charlie Kong, Hailiang Yu

**Affiliations:** 1College of Mechanical and Electrical Engineering, Central South University, Changsha 410083, China; waqas.farid111@gmail.com (W.F.); 213811001@csu.edu.cn (H.L.); 2Light Alloy Research Institute, Central South University, Changsha 410083, China; 3State Key Laboratory of Precision Manufacturing for Extreme Service Performance, Central South University, Changsha 410083, China; 4Shimadzu (China) Co., Ltd., Shanghai 200233, China; sshwzy@shimadzu.com.cn (Z.W.); cuihuijie1984@163.com (H.C.); 5Mark Wainwright Analytical Centre, University of New South Wales, Sydney, NSW 2052, Australia; c.kong@unsw.edu.au

**Keywords:** mechanical properties, finite element model, Al–TiC composite, RVE model, microstructure

## Abstract

This study investigates the mechanical properties of titanium carbide/aluminum metal matrix composites (AMMCs) using both experimental and computational methods. Through accumulative roll bonding (ARB) and cryorolling (CR) processes, AA1050 alloy surfaces were reinforced with TiCp particles to create the Al–TiCp composite. The experimental analysis shows significant improvements in tensile strength, yield strength, elastic modulus, and hardness. The finite element analysis (FEA) simulations, particularly the microstructural modeling of RVE−1 (the experimental case model), align closely with the experimental results observed through scanning electron microscopy (SEM). This validation underscores the accuracy of the computational models in predicting the mechanical behavior under identical experimental conditions. The simulated elastic modulus deviates by 5.49% from the experimental value, while the tensile strength shows a 6.81% difference. Additionally, the simulated yield strength indicates a 2.85% deviation. The simulation data provide insights into the microstructural behavior, stress distribution, and particle–matrix interactions, facilitating the design optimization for enhanced performance. The study also explores the influence of particle shapes and sizes through Representative Volume Element (RVE) models, highlighting nuanced effects on stress–strain behavior. The microstructural evolution is examined via transmission electron microscopy (TEM), revealing insights regarding grain refinement. These findings demonstrate the potential of Al–TiCp composites for lightweight applications.

## 1. Introduction

Aluminum matrix composites (AMCs) have garnered significant attention in the modern era due to their exceptional properties in terms of high strength, stiffness, and wear resistance, making them highly desirable for use in engineering applications. The incorporation of reinforcing ceramic particles into the metal matrix has been proven to further enhance these desirable properties. Among the various ceramic particles, TiC has emerged as a particularly promising candidate due to its remarkable and exceptional mechanical properties [1,2].

To fabricate AMCs, accumulative roll bonding (ARB) and cryorolling (CR) have become two prominent processing techniques. ARB entails the repeated rolling of material sheets, while CR follows a similar process at cryogenic temperatures. Both techniques result in the formation of ultrafine-grained microstructures within the AMCs, which contribute to improved mechanical properties [3]. Several studies have investigated the application of ARB for fabricating metal matrix composites (MMCs). Baazamat et al. [4] explored the impact of ARB cycles on the properties of aluminum-based composite-reinforced nanoparticles. The ARB led to homogeneous ceramic particles’ distribution within the matrix; this leads to improved tensile strength and hardness. In research conducted by Yu et al. [5], CR was employed to fabricate nano-structural aluminum sheets. The CR induced the refinement of the microstructure, leading to enhanced mechanical characteristics, such as increased yield strength and improved wear resistance. Furthermore, the combination of ARB and CR has shown great promise in producing MMCs with exceptional properties [6]. Gao et al. [7] investigated the microstructural evolution and mechanical behavior of lightweight metallic laminated composites subjected to ARB followed by CR. The findings showed a notable decrease in grain size and rise in the composite’s yield strength. The use of ultrafine-grained microstructures, achieved through ARB and CR, has proven to be a key factor in enhancing the properties of MMCs.

Finite element analysis (FEA) has been used for predicting and understanding the mechanical behavior of AMCs. By simulating the composite’s response under tensile loading conditions, FEA provides valuable insights into the microstructural features that govern its mechanical properties. Numerous studies have explored the application of FEA in the analysis of AMCs. Kladovasilakis et al. [8] conducted a comprehensive FEA study to investigate the mechanical properties of additive composites. The simulation provided a detailed understanding of the stress distribution and deformation behavior, and it clarified graphene’s contribution to improving the composite’s mechanical performance. Similarly, Rogala et al. [9] employed FEA to study the stress–strain behavior of aluminum–silicon carbide composites. The simulation allowed them to analyze the influence of the particle size and distribution on the mechanical properties, aiding in the optimization of the composite’s performance for specific applications. FEA has also been applied to study the effects of different processing techniques on the microstructural behavior of AMCs. In a study by Graça et al. [10], FEA was used to analyze the impact of different rolling processes on the mechanical properties of aluminum–nanoceramic composites. The simulations provided valuable information on the distribution of stresses and strains, revealing a correlation between the microstructural features and the mechanical performance. The effectiveness of FEA in predicting the mechanical behavior of AMCs has also been validated by comparing simulation results with experimental data. Ohguchi et al. [11] conducted FEA simulations on aluminum composites and compared the predicted stress–strain curves with the experimental results. The close agreement between the simulation and experimental data demonstrated the accuracy and reliability of FEA as a predictive tool for AMCs. 

While a few studies [12,13,14] have explored the properties of AMCs using experimental and numerical approaches, a knowledge gap still exists. This paper aims to address this gap by investigating the influence of ARB and CR processes on the mechanical properties of an Al–TiCp composite, utilizing a combination of experimental characterization and numerical analysis via FEA. FEA has proven to be instrumental in studying AMCs reinforced with various particles, allowing researchers to explore the impact of the particle distribution and size on the materials’ properties [15,16]. Additionally, FEA has been utilized to investigate the effects of diverse processing techniques on the microstructure and mechanical behavior of AMCs [17]. The literature review showcases the wide-ranging applications of FEA in analyzing different types of AMCs and highlights its significance in uncovering the microstructural features that govern the composite’s mechanical properties. As FEA continues to advance, it is expected to play an even more integral role in the design and optimization of advanced AMCs for diverse engineering applications.

The novelty of this research lies in its integrated approach, which combines experimental analysis and finite element modeling to comprehensively investigate the mechanical properties of AMMCs. By leveraging advanced experimental fabrication techniques, alongside rigorous FEA methods in Abaqus, this study advances our understanding of composite behavior. Furthermore, our study specifically investigates the effect of different types of nanoparticles on the mechanical behavior of AMMCs. By systematically analyzing the impact of various particle shapes and sizes through computational simulations, we offer insights into the nuanced effects of these factors on a material’s mechanical properties. Our study incorporates mechanism-based modeling techniques to elucidate the underlying mechanisms governing the material’s behavior. By explicitly accounting for microstructural features and particle–matrix interactions, we can accurately predict the material’s response under loading conditions. This approach allows us to optimize the design of AMMCs for enhanced performance in specific applications.

## 2. Materials and Methods

In this study, the matrix material utilized was commercial pure aluminum 1050. The produced composites consisted of AA1050 alloy with sheet form measuring 200 × 100 × 1 mm, and had the chemical composition as presented in Table 1. To alleviate thermal stresses and prevent cracking, the sheets underwent annealing at 623 K (1 h) and were subsequently cooled to ambient temperature prior to the ARB process. The reinforcement used in this study was TiC powders with an average size of 2.5 µm as shown in Figure 1. OM was used to calculate the average grain size of annealed aluminum as shown in Figure 2. 

The experiments of this study involved systematic dual-stage technique for the production of AMCs. The objective was to fabricate high-quality Al–TiCp composite with enhanced mechanical properties. The entire process was carried out with utmost precision to ensure reproducibility and accuracy in the results. The ARB process commenced with the preparation of three AA1050 sheets, each measuring 0.80 mm in thickness. To promote effective bonding and improve surface roughness, the aluminum sheets were subjected to meticulous wire-brushing procedure to reduce surface oxide layers. Subsequently, TiC particles were judiciously dispersed between the sheets before initiating the ARB process. In the initial ARB cycle, 2.0 weight percent of TiC particles were incorporated to reinforce the composite, while subsequent cycles were performed without the addition of any particles to establish controlled comparison. The weight percentage of TiC particles was calculated based on the total weight of the three aluminum sheets used in the composite fabrication process. The total weight of these aluminum sheets was measured to be 88.79 g. From this total weight, the 2.0 weight percent of TiC particles was determined, amounting to 1.78 g.

The ARB cycle was initiated without the use of any lubrication, and the sheets were preheated at 623 K for 5 min to optimize the bonding process. The rolling reduction during the ARB cycle was set at 55%, aiming to achieve the desired grain refinement and homogeneity within the composite. After each ARB cycle, the sheets underwent cutting, surface preparation, stacking, and roll bonding, ensuring comprehensive integration of the TiC particles into the aluminum matrix. To incorporate the TiC particles into the composite during the ARB process, a meticulous approach was followed. The TiC particles were poured through 500 mesh sieve and gently sprinkled on sheet to ensure uniform distribution and avoid clustering of particles. Following the successful ARB process, the resulting sheet was subjected to CR at 77 K. The CR process was conducted for a maximum of five cycles, with each cycle involving a 26% reduction in thickness. As a result of the CR, the thickness diminished to an optimized value of 0.15 mm, leading to further enhancement of the composite’s mechanical properties.

The tensile strength of AMCs was assessed using Shimadzu (Kyoto, Japan) AGS-X 10-KN tensile tester, operating at strain rate and testing speed of 1.0 mm/min. The tensile tests were conducted at room temperature. Micro-hardness was assessed using an HXD-2000TMC/LCD 181101X Vickers tester (London, UK), applying 10 g load for 10 s on various cycle samples. To analyze composite fracture morphology and microstructure, a TESCAN (Brno, Czech Republic) MIRA3 LMU TEM and SEM were employed.

## 3. Results

### 3.1. Microstructure

The experimental study utilized SEM to examine the microstructure of the Al/TiC MMCs during various rolling steps. Figure 3 depicts the SEM images of the Al/TiC composites. In Figure 3a, the TiC particle layers can be observed between the aluminum after ARB-1. The TiC particles initially aggregated in the early ARB stages, leading to the creation of TiC particle-free zones in all the samples. However, with increasing cycles, the TiC particle clustering decreased. The implementation of cryorolling after the ARBed process led to the transformation of the laminated shape composite, resulting in a uniformly distributed particle composite (Figure 3d) compared to the other cycles, although some unreinforced particles were also observed. Guo et al. [18] used SEM to characterize the microstructure and composition of silica–aluminum composite powders obtained through ball milling. Hullur et al. also [19] conducted a microstructural analysis of the AMCs reinforced with ceramic particles using SEM. They observed a uniform dispersion of the ceramic particles within the aluminum matrix, which significantly contributed to the material’s mechanical strength and deformation behavior. The materials were subjected to an XRD examination set at a rate of 0.45 min^−1^.

### 3.2. Modeling and Simulation

The integration of this analysis with the models contributes to a more comprehensive evaluation of the composite’s behavior under tensile loading conditions. The insights gained from these results serve as a foundation for further investigations and design considerations regarding the development of advanced MMCs. The FEM of the Al–TiCp composite was meticulously generated utilizing the Abaqus software. To emulate the experimental conditions, the model was subjected to tensile loading with fixed support applied on the back side. The computational outcomes were then compared to the experimental data to confirm the numerical model’s accuracy.

The generation of the finite element model is a crucial step in investigating the composite microstructures in terms of deformation evolution. To achieve this, the deliberate manipulation of the material’s microstructure is carried out, involving the incorporation of distinct regions representing the matrix and reinforcement particles. By extracting the X–Y coordinates of all the vertices, the segmented image is effectively transformed into the computational finite element domain. Figure 4 portrays the outcome of the segmentation process conducted using Abaqus, showcasing the computational domain established to mimic the real composite microstructure. As shown in Figure 4a, the SEM image of the actual composite microstructure serves as the reference for creating the segmented microstructure, as illustrated in Figure 4b. This segmented microstructure allows for the precise representation of the reinforcement particle and matrix regions within the model. In a study by Gao et al. [20], they utilized FEA and the Abaqus software to generate a segmented microstructure of AMCs reinforced with ceramic particles. In Figure 4c, the mesh used in the RVE−1 model is illustrated, comprising 18,117 nodes and 36,152 linear triangular elements of type CPS3. By analyzing the behavior of the material subjected to tension through the finite element analysis of the model, the critical mechanical characteristics, such as strength, deformation behavior, and stress distribution, can be accurately evaluated. The utilization of the finite element mesh with a substantial number of nodes and elements ensures a highly detailed and accurate representation of the material’s microstructure. This level of refinement allows for the precise examination of the stress concentrations, deformation patterns, and failure mechanisms within the composite material. The finite element model, as depicted in Figure 4d, was constructed with fixed constraints applied to the edges of the model, while an external tensile load was simulated by applying forces to pull the model in opposite directions. The fixed constraints on the edges of the RVE−1 model ensure that the boundaries remain immobile, reflecting a realistic scenario where the material is anchored or clamped during testing. The applied tensile load allows for the simulation of the composite material’s behavior under tension, providing crucial insights into its mechanical response. Through this advanced microstructural modeling and finite element analysis approach, the study aims to gain comprehensive insights into the mechanical characteristics of the matrix.

By employing this sophisticated microstructural modeling technique, the computational RVE−1 model effectively captures the key features and distribution of the particles within the composite material. Simulating the deformation and damage evolution of the composite with this microstructure model provides its mechanical behavior under tensile loading. The application of Abaqus in this microstructural modeling process [21,22,23,24,25] ensures the accurate representation and alignment of the reinforcement particles, paving the way for the reliable investigation into the influence of the microstructural variations on the overall mechanical response of the composite material. Table 2 provides the essential material parameters for both the TiC particles and the Al matrix.

In Figure 5, an exploration into the microstructural modeling with varying shapes and sizes of particles is presented. Indeed, RVE−2 (small-sized (V_f_ = 2%) circular TiC particles), RVE−3 (large-sized (V_f_ = 8%) TiC circular particles), RVE−4 (small-sized (V_f_ = 2%) TiC triangular particles), and RVE−5 (large-sized (V_f_ = 8%) triangular particles) are depicted in Figure 5a,c,e,g, respectively, showcasing the detailed mesh configurations in Abaqus. Figure 5b,d,f,h illustrate the corresponding constraints, emphasizing a fixed back and an applied tension load on the opposite side. This configuration is essential for simulating the realistic mechanical loading conditions regarding these models. These constraints ensure that the simulation mirrors the mechanical loading conditions encountered by the composite material in practical applications. By creating distinct models with variations in the particle shape and size, this study delves into the nuanced effects of the microstructural configurations on the mechanical behavior of the composite material. These detailed models and constraints enable a comprehensive analysis, shedding light on the particle geometry in terms of influencing the overall efficacy of the matrix.

In essence, Figure 5 showcases the intricacies of the microstructural modeling, emphasizing the importance of considering the particle morphology in predicting the mechanical behavior of MMCs. This approach revealed the manner in which different sizes and shapes regarding the particle characteristics at the microscale translate into macroscopic mechanical responses, guiding the optimization of the composite materials for specific engineering applications.

The interface model is developed to account for the interactions and bonding between the two constituents, thereby capturing the synergistic effects that arise from their combination. The interface bonding strength is represented by an interface strength parameter (σ_interface). It quantifies the ability of the aluminum matrix and titanium carbide reinforcement to adhere and transfer the load effectively across the interface. The value of the σ_interface is determined either through experimental measurements or is obtained from previous studies of similar composite systems. To simulate the behavior of the metal, a cohesive zone model is employed. The cohesive zone model incorporates cohesive elements along the interface region, where the interface strength parameter (σ_interface) is applied. The cohesive elements enable the modeling of the interface’s deformation and damage, enabling the study of the crack initiation, propagation, and cohesive failure within the interface. The cohesive zone model is represented by the traction–separation law, which defines the relationship between the cohesive stress (τ_coh) and the separation (δ) between the interface surfaces. This can be expressed as
τ_coh = f(δ),(1)
where f(δ) is the cohesive traction–separation law that characterizes the softening behavior of the interface as δ increases.

The cohesive zone model defines the softening response, representing the damage initiation and propagation within the interface. As the applied stress surpasses the interface strength parameter (σ_interface), the cohesive elements begin to soften, signifying the debonding and damage initiation. The evolution of the interface damage is tracked as the composite undergoes mechanical loading, providing insights into the failure mechanisms and behavior at the interface. The damage initiation at the interface is governed by the damage parameter (D), which evolves based on the cohesive zone model. As the cohesive stress exceeds the interface strength (σ_interface), the damage initiates and evolves over the interface. This can be described as
D = g (τ_coh − σ_interface),(2)
where g is damage evolution function that depends on the difference between the cohesive stress (τ_coh) and the interface strength (σ_interface).

The cohesive elements interact with the surrounding finite elements representing the aluminum matrix and titanium carbide reinforcement. The interaction between the cohesive elements and bulk elements allows for the load transfer between the constituents, reflecting the bonding and stress distribution at the interface. The cohesive elements interact with the surrounding finite elements, representing the aluminum matrix and titanium carbide reinforcement. The cohesive traction (τ_coh) is applied to the corresponding finite elements, representing the load transfer at the interface.

The cohesive zone model enables the calculation of energy dissipation due to the damage evolution at the interface. The energy dissipation quantifies the amount of energy absorbed during the crack propagation and cohesive failure, offering an assessment of the composite’s fracture toughness and resistance to interface failure. The energy dissipation (G_diss) due to interface damage evolution can be calculated as the area under the traction–separation curve within the cohesive zone. This can be expressed as
G_diss = ∫ τ_coh dδ,(3)
where the integral is taken over the cohesive zone as the interface undergoes deformation. The interface model is validated through comparisons with experimental data and observations of interface behavior. The model’s ability to predict interface debonding, damage initiation, and failure is verified against experimental results to ensure its accuracy and reliability. By incorporating the interface model into the overall FEM, the study gains the mechanical behavior of titanium carbide/aluminum metal matrix composites. The interface model is instrumental in elucidating the critical role of the interface in determining the composite’s strength, toughness, and overall performance under loading conditions.

### 3.3. Experimental Case (RVE−1)

Figure 6 presents the experimentally similar case of the TiC particles in the Abaqus simulation model (RVE−1), including σ_vm_, σ_xx_, σ_yy_, σ_xy_, and ε_max_. These results offer critical insights into the mechanical response of the TiC-reinforced aluminum composite. In Figure 6a, the highest σ_vm_ is reported as 432 MPa around the nanoparticles. This value represents the overall equivalent stress, taking into account both the normal and shear stresses. The distribution of σ_vm_ across the model is visualized, highlighting the regions of localized stress concentration, particularly around the TiC particles. The average peak value of 279 MPa (Figure 12) signifies the areas where the material experiences the highest combined stresses, providing essential information about the potential failure points and load-bearing capacities. Figure 6b through Figure 6d further dissect the stress components. The normal stresses in the x and y directions denoting σ_xx_ and σ_yy_ are depicted, respectively. These stress distributions showcase the material’s response to applied loads, with areas of tension. The interplay between the reinforcing TiC particles and the aluminum matrix is evident, influencing the stress distribution across the composite. Figure 6d details the shear stress (σ_xy_), revealing the distribution of the forces acting parallel to the material’s plane. Lastly, the color variation in ε_max_ (maximum and minimum strain values) illustrates the extent of the deformation, with higher values indicating regions experiencing greater strain in Figure 6e. In the research by Mangal et al. [26] and Wu et al. [27], they utilized FEA to investigate the distribution of the maximum plastic strain within the FEA of a matrix under tensile loading. The FEA model represents a small volume of the composite material [28,29], which is used as a representative sample to study its behavior on a microscale level.

### 3.4. Effect of Particle Size and Shape

In this Abaqus-constructed composite material comprising aluminum reinforced with TiC particles, subjected to a uniform load along the x-axis with fixed boundaries on opposite sides, we explored the impact of the particle size and shape on the mechanical behavior. In Figure 7, depicting the RVE−2 case model, the highest von Mises stress of 226 MPa around the particles was observed (Figure 7a). The area of the dark green color on the particles signifies maximum stress. Figure 8a, employing large circular particles (RVE−3), reveals a value of σ_vm_ of 227 MPa, with a distinct blue and green color pattern on the particles showing the maximum and minimum stress area. Transitioning to Figure 9a, featuring the RVE−4 case model, a value of σ_vm_ of 486 MPa is showcased. This underscores the superior mechanical properties of the smaller triangular particles, supporting the notion that the triangular shape may offer improved load transfer characteristics. In Figure 10a, illustrating the RVE−5 case model, a value of σ_vm_ of 502 MPa is observed. The colors on the particles and the aluminum matrix indicate the stress variation. Concurrently, the light blue and green areas on the aluminum matrix signify different stress distributions, emphasizing the interplay between the reinforcing particles and the adjacent matrix. The discernible increase in the mechanical properties in the larger and triangular particles further supports the idea that the unique geometry of the triangular particles contributes to the enhanced load-bearing capacity. The analysis of the composite material, particularly the mechanical response and stress distribution, unequivocally indicates the superior effectiveness of the RVE−5 case model. The discernible peaks in the stress concentrations around these particles signify a robust load-bearing capacity, showcasing their heightened effectiveness in reinforcing the aluminum matrix. This finding underscores the significance of the size of the particles in determining the mechanical strength of the matrix, advocating for the strategic utilization of the RVE−5 case model in the design and optimization of advanced materials for superior performance and application-specific requirements. The purpose of utilizing RVE−2, RVE−3, RVE−4, and RVE−5 is to explore the effectiveness of various particle shapes, particularly irregular shapes compared to circular ones, in enhancing the mechanical properties of the composite material. By employing different RVE models with distinct particle shapes while maintaining the same volume percentage, we aim to investigate how irregular shapes, such as triangular particles, contribute to improving the material’s performance. Irregular shapes tend to have sharper edges and points, which can embed into the matrix more effectively.

In Figure 11, the graphical representation of the variations in the equivalent and normal residual stresses provide crucial insights into the mechanical response of the composite material, specifically focusing on σ_vm_, σ_xx_, σ_yy_, and σ_xy_. The graph showcases a distinctive sine wave pattern, with the peaks corresponding to the TiC particle regions and the troughs aligning with the aluminum matrix areas across all the models. This sinusoidal behavior suggests a cyclical variation in the stresses within the composite material, with notable peaks around the TiC particles and lower points in the aluminum matrix regions. We observed varying levels of σ_mises_ across the different regions of the composite material, as evidenced by the average and highest peak values recorded for each model. For RVE−2, RVE−3, RVE−4, and RVE−5, the average peak values of σ_mises_ were measured at 131 MPa, 141 MPa, 213 MPa, and 257 MPa, respectively, with the corresponding highest peak values of 175 MPa, 198 MPa, 438 MPa, and 475 MPa. These findings shed light on the areas within the composite material experiencing the highest combined stresses, particularly at the interfaces between the aluminum matrix and TiC particles. The cyclic fluctuation in the overall equivalent stress (σ_vm_), depicted in Figure 11a, highlights the localized stress concentrations around the TiC particles, with the troughs indicating the regions of reduced stress in the aluminum matrix. Similarly, Figure 11b,c illustrate the normal stresses (σ_xx_ and σ_yy_) in the x- and y directions, respectively, showing elevated stress levels around the TiC particles and lower stress concentrations in the aluminum matrix regions. Furthermore, the variations in shear stress (σ_xy_), as shown in Figure 11d, underscore the interaction between the aluminum matrix and TiC particles, with the peaks indicating increased shear stress around the TiC particles. This comprehensive analysis provides essential information regarding the potential failure points and load-bearing capacities within the composite material, informing the design optimization efforts aimed at enhancing the mechanical properties and reliability. The peaks around the TiC particles signify increased shear stress, while the troughs in the aluminum matrix regions indicate lower shear stress. The sinusoidal variation observed in Figure 11 suggests a cyclic redistribution of the stresses within the composite material, emphasizing the localized concentration around the TiC particles. This cyclic behavior is attributed to the interaction and load transfer dynamics between the reinforcing TiC particles and the aluminum matrix.

Figure 12 provides comprehensive insights into the stress–strain behavior of the Al–TiC composite under tensile loading, elucidated through the finite element analysis in Abaqus. Figure 12 encapsulates the stress–strain curves for RVE−1 through RVE−5. The experimental case (RVE−1) tensile stress value observed is 279 MPa, as shown in Figure 12. The distinctive curve showcases the diverse mechanical responses of the composite models with varying shapes and sizes regarding TiC reinforcement. Notably, the RVE−5 case model stands out with a tensile stress of 369 MPa, significantly surpassing the values for RVE−2 (139 MPa), RVE−3 (179 MPa), and RVE−4 (315 MPa). The significantly higher tensile stress exhibited by the RVE−5 case model reinforces the notion that large-sized triangular particles contribute notably to the overall strength of the Al–TiC composite.

The prediction of the modulus (E_comp) curve model involves considering the contributions from the aluminum matrix and the titanium carbide reinforcement. Similar to the stress–strain curve prediction, the Young’s modulus of the composite can be determined using the rule of mixtures, which takes into account the volume fractions and the Young’s moduli of the individual constituents. The rule of mixtures for predicting the modulus of composite is provided by
1/E_comp = V_al/E_al + V_TiC/E_TiC,(4)
where
E_comp is the overall Young’s modulus of the composite;E_al is the modulus of the aluminum matrix;E_TiC is the modulus of the titanium carbide reinforcement;V_al is the volume fraction of the aluminum matrix in the composite; andV_TiC is the volume fraction of the titanium carbide reinforcement in the composite.

The modulus of the aluminum matrix (E_al) and the titanium carbide reinforcement (E_TiC) can be obtained from experimental testing or appropriate material models that represent their respective mechanical behaviors. Using the rule of mixtures, the overall Young’s modulus of the composite can be predicted as a function of the volume fractions and Young’s moduli of the constituents. This predicted Young’s modulus curve provides valuable insights into the composite material’s stiffness and elastic behavior under tensile loading conditions. As with the stress–strain curve, the predicted Young’s modulus curve can be compared with the experimental findings to validate the accuracy of the composite model.

In Figure 13, the Young’s moduli of the Al–TiC composite models (RVE−1 to RVE−5 case models) are comprehensively examined through the simulation results and mixture rule formula (Equation (4)). The simulation-derived Young’s modulus values for each model are specified as follows: RVE−1 (91 GPa), RVE−2 (81 GPa), RVE−3 (84 GPa), RVE−4 (88 GPa), and RVE−5 (104 GPa). Correspondingly, the Young’s modulus values obtained through the mixture rule formula for each model are as follows: RVE−1 (85 GPa), RVE−2 (81 GPa), RVE−3 (82 GPa), RVE−4 (86 GPa), and RVE−5 (107 GPa). This detailed array of values provides profound insights into the distinct mechanical responses influenced by the various shapes and sizes of the TiC reinforcement within the aluminum matrix, further contributing to a nuanced understanding of the composite material’s behavior. Moreover, the comparison between the simulated and formula-derived Young’s modulus values highlights the efficacy of the simulation approach in predicting the composite material’s mechanical behavior. Notably, the simulation results align closely with the values obtained through the mixture rule formula, affirming the reliability of the finite element analysis in capturing the complex interactions between the aluminum matrix and TiC reinforcement. The observed variations in the Young’s modulus values among the different models underscore the nuanced influence of the particle size and shape on the composite’s stiffness. The previous studies provide a valuable foundation for the current analysis, offering insights into the key factors influencing the prediction of the stress–strain and Young’s modulus curves for the composites [30,31]. By evaluating the contributions of the aluminum matrix and titanium carbide reinforcement, this investigation serves to offer insightful information regarding the composite material’s overall mechanical properties. The stress–strain curve of the matrix model involves considering the contributions from both the aluminum matrix and the titanium carbide reinforcement. The overall stress–strain behavior of the composite can be determined by combining the stress–strain responses of the individual constituents using appropriate mixing rules. One commonly used mixing rule for predicting the stress–strain curve of the matrix is the rule of mixtures.

### 3.5. AMCs’ Microstructural and Mechanical Properties of Al–TiCp

#### 3.5.1. Microstructural Evolution

The current investigation utilized XRD to ascertain the constituent phases within the composite material under examination. XRD was employed to meticulously analyze the samples, aiming to elucidate the variations in the dislocation density and crystallite size induced by the ARB and CR processes. The XRD analysis was conducted using a Bourenvestink DRON-8 X-ray diffractometer, renowned for its precision and reliability in characterizing crystalline materials. Operating the XRD apparatus at the tube voltage of 40 kV and current of 20 mA ensured optimal X-ray generation conditions, facilitating robust and reliable data acquisition. The utilization of the Cu-Kα target further enhanced the accuracy and sensitivity of the analysis given its favorable wavelength for probing crystalline structures. Figure 14 shows the XRD patterns representing the profiles of the composite material formed via the specific fabrication processes, and the compacted initial Al. This finding reinforces our achievement in fabricating AMCs with reinforcement. The peaks that are observed are only related to the TiC, and no extra phase is detected. The intensity and width of the peaks seem to be subject to change as a result of the ARB and CR processing, which can be attributed to the changes in the crystal orientation, substructure size, dislocation density, and microstrains. Notably, the lack of detrimental phases formed during the fabrication is emphasized, underscoring the material’s integrity. The Al-SiCp composites were subjected to an XRD investigation by Kim et al. [32], who identified the phases produced by the microwave and conventional heating methods. The analysis showed the peaks of the aluminum matrix, along with distinct peaks indicating the presence of SiCp particles.

The specific features were captured in the TEM images. Figure 15a–f provide a detailed examination of the microstructural evolution, showcasing the dislocation cells, dislocation lines, tangled dislocations, microvoids, and subgrains after the CR-5 cycle. The microstructural insights derived from these images reveal crucial details about the material’s characteristics. The incorporation of the TiC particles into the aluminum matrix is evident, leading to a notable increase in the dislocation density. This rise in the dislocation density is primarily concentrated at the interface between the TiC particles and the matrix, serving to accommodate the strain incompatibility between the two phases. This cycle’s microstructure shows the regions with dislocation tangles and a mixture of deformed and undeformed grains. With the progression of the cycles, there is a simultaneous increase in the dislocation density and a reduction in the subgrain density. The movement of the dislocations toward the boundaries results in a decrease in the dislocation density within the cells. Notably, the grains situated in close proximity to the TiC particles exhibit distinct orientations in comparison to the grains in the other regions. This phenomenon is primarily attributed to the significant local lattice rotations induced by the undeformability of the TiC particles during the ARB process. The different grain alignments near the TiC particles contribute to the observed microstructural heterogeneity within the composite specimen. Lu et al. [33], in their study, investigated the microstructural evolution of Al with TiC composites during severe plastic deformation. They observed a significant increase in the dislocation density at the interface between the Al matrix and TiC particles, leading to the formation of dislocation tangles. This study also highlighted a refinement in the subgrain size and the presence of ultrafine-grained structures as the deformation cycles progressed [33].

#### 3.5.2. Mechanical Properties

In Figure 16a, the stress–strain curves and hardness values of the as-received aluminum and Al–TiC composites produced via the ARB + CR method are presented. Following annealing, the composite exhibited the highest tensile strength at 261 MPa, indicating a 3.6 times increase compared to the as-received material. According to previous research, the primary strengthening mechanisms are responsible for the strength increase in extensively deformed materials [34,35]. Notably, the annealed composite consistently outperformed the ARB composite samples regarding the tensile strength throughout the CR cycles, showcasing a continual enhancing trend (Figure 16a). Moreover, the incorporation of the TiCp ceramic particles into the aluminum matrix notably bolstered the tensile strength, albeit at the expense of reduced elongation. This trend is linked to plastic instabilities, as past research indicates [36]. The prevalence and properties of these plastic instabilities depend on variables like each constituent’s strength coefficient, work-hardening exponent, and initial thickness [36]. The micro-hardness results (Figure 16b) demonstrated a rising trend with the number of ARB and cryorolling cycles in the Al–TiC composite. The composite’s micro-hardness showed rapid escalation compared to the annealed aluminum matrix, reaching a peak value of 84 Hv. The initial cycles displayed a high strain-hardening rate, while the composite’s ARB process indicated a comparatively gradual increase in hardness. The ceramic particles were crucial for the grain refinement and dislocation in the composite. The high strain-hardening rate that arises from their interactions with the dislocation density causes the increase in the micro-hardness in the early cycles [3]. Subsequent to the initial CR passes, the composite’s hardness experiences a rapid escalation, primarily due to the grain refinement [37]. These developments align with the strain-hardening and strengthening mechanisms observed in the context of the tensile properties. Moreover, the determination of the elastic modulus (Figure 16c) was accomplished using the graphical technique, revealing an approximate value of 70 GPa for the pure aluminum. Among the various ARB and cryorolling cycles, the composite at CR-5 exhibited the highest elastic modulus values, measuring 86 GPa (Figure 16c).

In Figure 17a, it is evident that the yield strength of the composite experiences a substantial increase compared to the as-received material. This notable enhancement can be primarily attributed to the processing techniques applied. In the ARB process’s early stages, the notable strain hardening results in an initial increase in the composite’s yield strength. This is supplemented by the effective incorporation and dispersion of the TiC particles within the aluminum matrix, contributing to the efficient load transfer mechanism and reinforcing the composite structure. The yield strength further improves with increasing rolling cycles, primarily due to the continual progressive strengthening of the composite material. This refinement is coupled with the establishment of stronger bonds between the layers and the reduction in any potential defects or porosities within the composite structure. Consequently, these effects lead to a notable increase in the yield strength of the Al–TiC matrix, demonstrating the effectiveness of the processing techniques in enhancing the material’s mechanical properties. As depicted in Figure 17b, the elongation exhibited a sharp decline from 21% (for the annealed raw material) to 4.5%, followed by a slight increase, and subsequently a decrease to 2.5% after eight cycles. The initial decrease in the ductility of the ARB-treated Al–TiCp composite sheet results from the significant strain hardening in the first cycle and the presence of porosities, particularly at the TiC particle–metal matrix interface, in the initial pass. The insufficient bonding at the interface and the accumulation of TiC particles between the layers also contribute to the decline in ductility during the initial passes. The rise in ductility during the ARB and CR cycles results mainly from the TiC particle dispersion, promoting excellent bonding in between the layers and reducing the porosities. Furthermore, as the cycles increase, the elements, including the finer grain size, the greater boundary misorientation, and the existence of free dislocations, may further improve the ductility [38]. Gatea et al. [38] studied the impact of the particle size and distribution on the mechanical properties of the Al/SiCp composites.

Figure 18 presents the microstructural fracture analysis of the composite samples at various processing stages. The SEM images show both lower- and higher-magnification views, offering insights into the failure mechanisms and bonding characteristics of the material. Figure 18a shows the fractured surface of the starting aluminum material without reinforced nanoparticles. In this case, the size and depth of the dimples are pronounced, indicating enhanced ductility. The presence of pronounced dimples influences the plastic deformation behavior, contributing to the enhanced ductile characteristics of the pure aluminum. On the other hand, Figure 18b exhibits the fractured surface of the final composite after undergoing the CR-5 cycles. This image demonstrates strong bonding in the composite, indicating the effective interfacial adhesion between the aluminum matrix and the TiCp reinforcement. The fractures in the final composite (Figure 18b) demonstrate crack conjunction, microvoid formation, and shear fracture. Such fracture patterns are typical in ductile materials and suggest that the composite material undergoes ductile rapture during failure. Furthermore, the dimple shape and orientation offer insights into the stress conditions and fracture propagation direction.

## 4. Discussion

The investigation into the mechanical properties of the Al–TiCp composite material through the combined experimental and numerical approach has provided valuable insights into the behavior and performance of the composite under tensile loading conditions. The comprehensive analysis of the data obtained from both the experimental testing and finite element simulations in Abaqus has contributed to a deeper understanding of the mechanical response and microstructural characteristics of the composite. The ARB and CR fabrication process significantly influences the microstructure of the TiCp-reinforced AA1050 composite surfaces. The resulting Al–TiCp composite exhibits notable improvements regarding the mechanical properties. The integration of the 2D micromechanical RVE−1 FEM in Abaqus enables the detailed exploration of the material’s mechanical behavior. The observed disparities between the experimental and computational results, although slight, underscore the inherent challenges associated with the manufacturing intricacies and the need for continual refinement regarding the computational models. In Figure 6, Figure 7, Figure 8, Figure 9 and Figure 10, the von Mises stresses and normal residual stress distributions provide vivid depictions of the stress concentration and distribution patterns within the composite materials. The discernible trends indicate that, for smaller particle sizes, triangular TiC particles exhibit superior mechanical properties, while, for larger sizes, triangular particles, particularly in the large size category, dominate in terms of strength and stress concentration. Figure 11 delves into the cyclic variation regarding the equivalent and normal residual stresses across the different models. The sinusoidal patterns observed further emphasize the dynamic redistribution of the stresses within the composite material. Notably, large-sized triangular TiC particles contribute to pronounced stress concentrations, affirming their efficacy in terms of load-bearing capabilities. The stress–strain curves in Figure 13 further reinforce the significance of the particle size and shape. This exceptional performance highlights the advantageous impact of larger triangular particles on the tensile behavior of the composite. The simulation results, particularly in Figure 13, provide valuable insights into the diverse mechanical responses exhibited by the different models. The close alignment between the simulated and formula-derived Young’s modulus values attests to the accuracy of the finite element analysis in capturing the complex interactions within the composite material. RVE−5 again emerges as the leader, underscoring the profound impact on enhancing the material’s rigidity. This comparison highlights the significance of the microstructural features in determining the composite material’s mechanical response and provides critical insights for optimizing designs based on specific size and shape configurations, advancing the materials’ engineering understanding. These studies are consistent with the existing body of literature on metal matrix composites, particularly the works of Zhuang et al. [39]. These studies highlighted the significance of the cohesive zone models in characterizing the interface behavior and damage evolution within the composite material. The incorporation of cohesive elements in the finite element analysis, as demonstrated in this research, enables a more comprehensive understanding of the crack initiation, propagation, and cohesive failure within the interface, aligning with the findings reported in the literature [40,41].

The TEM results revealed significant microstructural refinement and the presence of various key features, including microvoids, dislocation lines, tangled dislocations, dislocation cells, subgrains, and TiC particles. The presence of dislocation lines and dislocation tangles in the microstructure suggests the accumulation of dislocation densities, highlighting the influence of the processing techniques on the material’s deformation behavior [42]. Moreover, the TEM analysis vividly showcased the TiC particles’ contribution to elevating the dislocation density within the aluminum matrix. The documented rise in dislocation density and subsequent subgrain size reduction with escalating ARB and CR cycles highlight the processing methods’ efficacy in fostering the grain refinement and improving the material mechanical properties. The observed variations in the lattice orientations and the presence of distinct grain boundaries emphasize the development of a refined and intricate microstructure within the composite specimen. Additionally, the microstructural analysis of the tensile fracture surfaces revealed crucial insights into the failure mechanisms and bonding characteristics of the Al–TiCp composite. The observation of ductile rupture and the formation of dimples in both the reinforcement and the aluminum matrix signify the ductile behavior of the composite during failure, confirming the effective interfacial bonding between the constituents [43]. These results align with prior research by Wu et al. [44], highlighting the pivotal role of interfacial bonding in shaping the mechanical properties and overall performance of metal matrix composites. The integration of experimental analysis and finite element modeling has facilitated a comprehensive understanding of the Al–TiCp composite’s mechanical behavior and microstructural features. By leveraging the insights from previous studies [45,46], this research contributes to the broader understanding of composite materials and provides a foundation for further optimizing the design and performance of advanced metal matrix composites for various engineering applications.

Figure 19 illustrates this study’s exploration of the formation process and enhanced mechanical properties of the Al–TiCp composite, contrasting it with similar aluminum matrix systems reinforced with nanoparticles. Remarkably, the Al–TiCp composite exhibits notably superior tensile strength and yield strength compared to aluminum matrices reinforced using different techniques. In recent years, significant research has focused on understanding the intricate relationship between the mechanical and microstructure characteristics of composites [47]. Table 3 shows the comparison of the simulation and experimental results for the mechanical characteristics of composites. The experimental values, representing the elastic modulus, tensile strength, and yield strength, are measured as 86 GPa, 260 MPa, and 239 MPa, respectively. The simulation results for RVE−1 reveal close agreement with the experimental values but with notable discrepancies. The simulated elastic modulus is calculated as 91 GPa, exhibiting a 5.49% deviation from the experimental value. Similarly, the simulated tensile strength is 279 MPa, representing a 6.81% difference, while the simulated yield strength is 246 MPa, indicating a 2.85% deviation. The observed discrepancies between the experimental and computational results could be attributed to various factors, including the simplifications implemented in the computational model, assumptions regarding material behavior, and potential variations in the manufacturing process. It is noteworthy that, despite these discrepancies, the simulation results generally align well with the experimental values, and the calculated error percentages fall within the acceptable ranges for many engineering applications. The error percentages of 5.49%, 6.81%, and 2.85% for the elastic modulus, tensile strength, and yield strength, respectively, suggest that the computational model, particularly for RVE−1, provides reasonably accurate predictions regarding the material’s mechanical behavior. Despite the slight differences, the Abaqus-based RVE−1 models effectively predict the mechanical behavior of the Al–TiC composite, confirming the usefulness and precision of the simulation method.

The comparison between the experimental and RVE−1 model approaches reveals a small difference in the ultimate tensile strength values. The successful correlation between the experimental and modeling data reinforces the confidence in using finite element analysis to further investigate the composite material’s behavior under different loading conditions and optimize its mechanical properties for specific engineering applications. The study conducted by Han et al. [55] investigated the stress–strain curves and compared the results with the numerical simulations in Abaqus. The experimental stress–strain curve provided valuable insights into the composite’s mechanical response under tensile loading [56]. Abaqus, a finite element analysis software, was utilized to model the material’s behavior, enabling a comprehensive comparison between the experimental and simulated stress–strain curves [57] (Table 4).

In the study by Derakhshani et al. [58], they conducted experimental testing on composite material to determine its elastic modulus. The experimental-based approach involved subjecting the composite to tensile loading and measuring the resulting strain response. The elastic modulus, representing the stiffness, was derived from the stress–strain data obtained experimentally. The increased rolling pressure during ARB and cryorolling significantly reduced the porosity at the Al–TiCp interfaces, enhancing the bonding strength, thus improving the composite mechanical properties. Ductile rupture and dimple formation were the prominent failure mechanisms observed in both the TiCp reinforcement and fractured Al surfaces, which retained ductility. The pronounced dimples in the starting aluminum indicate increased ductility, while the strong bonding in the final composite confirms the effective interfacial adhesion [59]. Understanding these microstructural features is crucial for optimizing the manufacturing processes and designing composites with superior mechanical properties.

## 5. Conclusions

The Al–TiC composite was fabricated successfully. The addition of TiC particles in the aluminum improved the Al–TiCp composite properties. The experimental analysis revealed significant improvements in the elastic modulus, yield strength, and hardness of the composite. Moreover, the RVE−1 finite element modeling offered valuable insights into the Al–TiCp composite’s mechanical behavior, demonstrating good agreement with the experimental data. Furthermore, the simulation results provided insights into the microstructural behavior, including the stress distribution and particle–matrix interactions, enhancing the understanding of the material’s mechanical response. Comparing the simulation results with the experimental data validated the accuracy of the computational models, ensuring confidence in the study’s findings and conclusions. By simulating different scenarios, the results facilitated design optimization, identifying the optimal particle shapes and sizes for enhanced performance in specific applications. The simulation results revealed that larger TiC particles notably increased the strength, emphasizing the size’s impact, while the triangular shapes exhibited superior strength compared to their circular counterparts. The assessment of Young’s modulus in the Al–TiC composite models through the simulation and mixture rule formula demonstrated good accuracy, providing accurate insights into the diverse mechanical responses induced by the varying shapes and sizes of the TiC reinforcement within the aluminum matrix. Furthermore, the TEM images revealed a refined microstructure, enhancing the overall strength and hardness. The microstructural analysis of the tensile fracture surfaces indicated ductile rupture and effective interfacial bonding. However, it is important to acknowledge the limitations of this study. The current research primarily focused on a specific set of particle sizes and shapes, limiting the generalizability of the findings to broader material systems. Additionally, the simulations assumed idealized conditions and may not fully capture the complex real-world behaviors of the composite material. The future research directions could involve expanding the scope of the particle size and shape variations to explore a wider design space. Furthermore, the experimental validation of the simulation predictions under more diverse loading conditions would enhance the reliability of the computational models. Additionally, investigating alternative reinforcement materials and exploring advanced manufacturing techniques could lead to further improvements in the composite performance and functionality.

## Figures and Tables

**Figure 1 materials-17-02093-f001:**
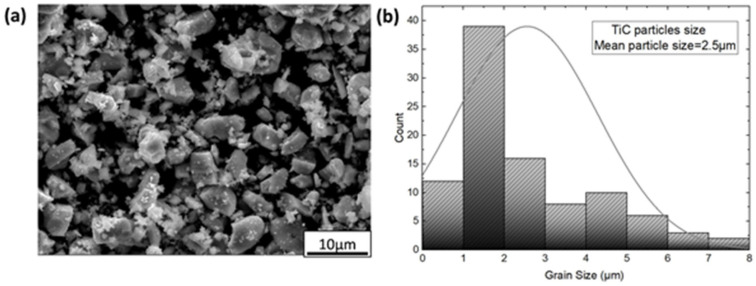
Diagrams of (**a**) SEM TiC particles and (**b**) particle size, respectively.

**Figure 2 materials-17-02093-f002:**
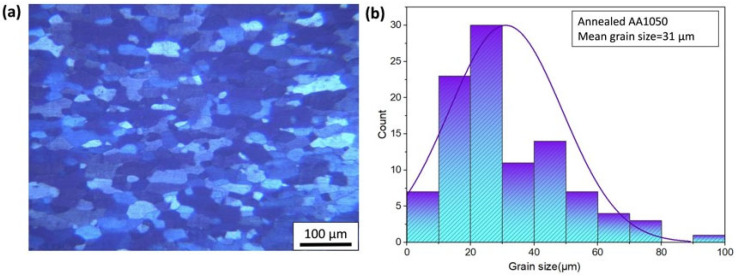
Diagrams of (**a**) OM annealed aluminum and (**b**) grain size.

**Figure 3 materials-17-02093-f003:**
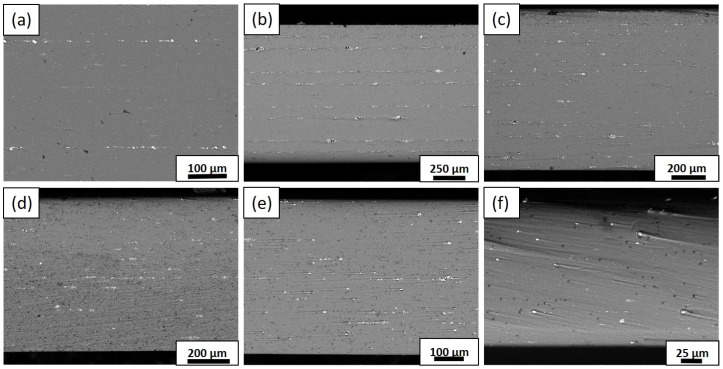
SEM micrographs of Al–TiC composites: (**a**) ARB-1, (**b**) ARB-3, (**c**) ARB-5, (**d**) CR-1, (**e**) CR-3, and (**f**) CR-5.

**Figure 4 materials-17-02093-f004:**
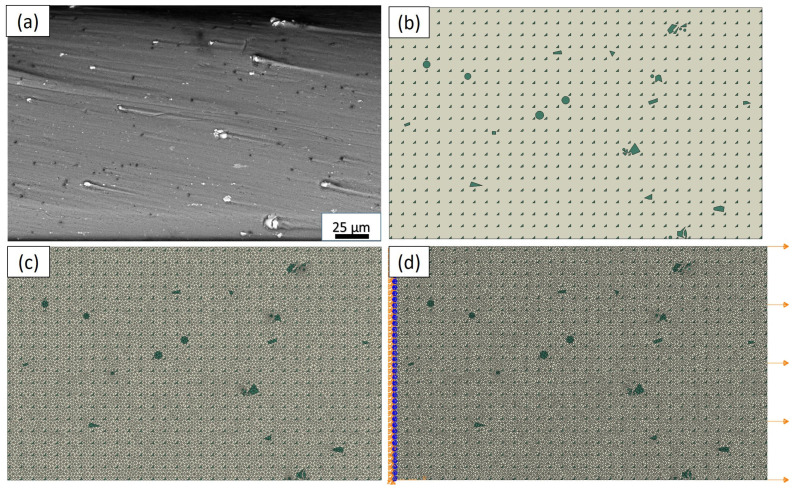
Microstructural modeling of RVE−1 (**a**) SEM real composite microstructure; (**b**) segmented microstructure (**c**) mesh, and (**d**) constraints.

**Figure 5 materials-17-02093-f005:**
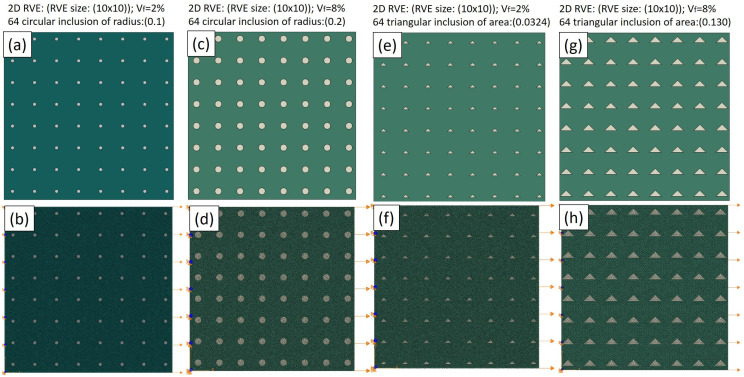
Microstructural modeling of RVE−2−5 (**a**,**c**,**e**,**g**) mesh and (**b**,**d**,**f**,**h**) constraints, respectively.

**Figure 6 materials-17-02093-f006:**
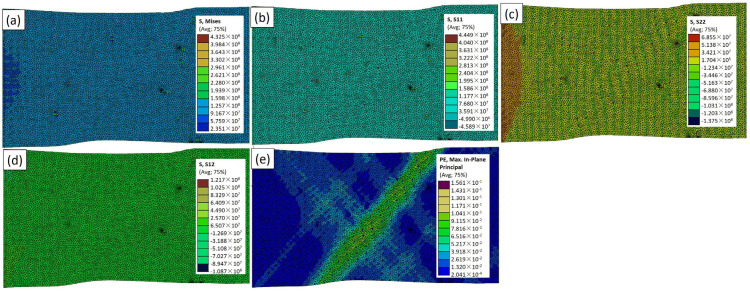
Simulation results for RVE−1 case: (**a**) σ_vm_; (**b**) σ_xx_; (**c**) σ_yy_; (**d**) σ_xy_; and (**e**) ε_max_ distributions.

**Figure 7 materials-17-02093-f007:**
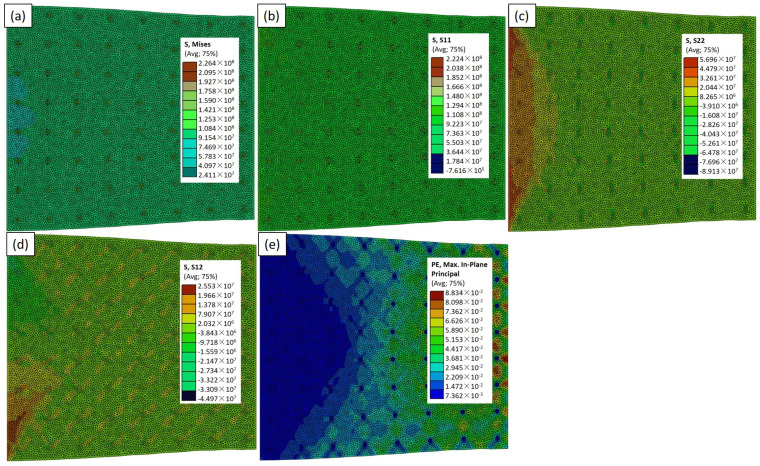
Simulation results for RVE−2 case: (**a**) σ_vm_; (**b**) σ_xx_; (**c**) σ_yy_; (**d**) σ_xy;_ and (**e**) ε_max_ distributions.

**Figure 8 materials-17-02093-f008:**
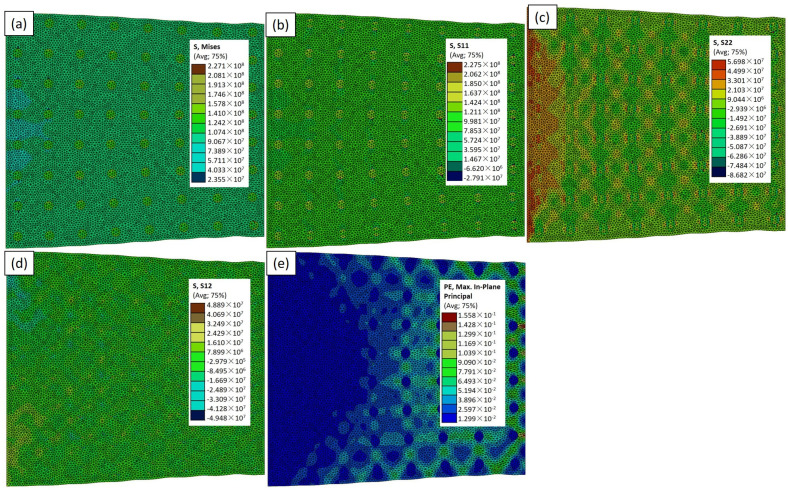
Simulation results for RVE−3 case: (**a**) σ_vm_; (**b**) σ_xx_; (**c**) σ_yy_; (**d**) σ_xy_; and (**e**) ε_max_ distributions.

**Figure 9 materials-17-02093-f009:**
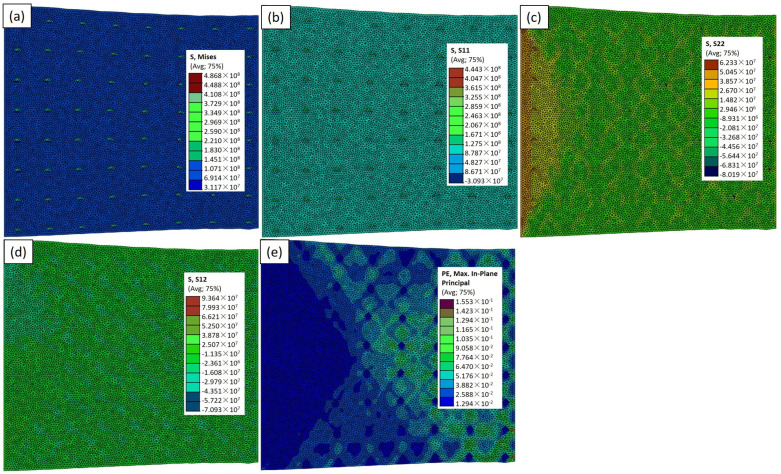
Simulation results for RVE−4 case: (**a**) σ_vm_; (**b**) σ_xx_; (**c**) σ_yy_; (**d**) σ_xy_; and (**e**) ε_max_ distributions.

**Figure 10 materials-17-02093-f010:**
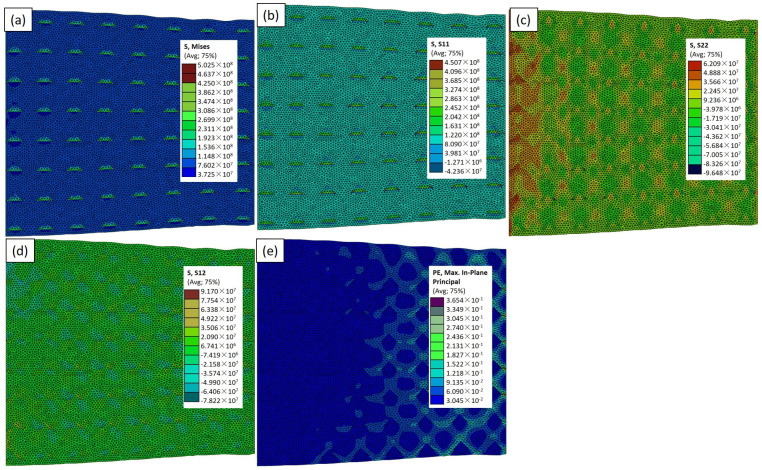
Simulation results for RVE−5 case: (**a**) σ_vm_; (**b**) σ_xx_; (**c**) σ_yy_; (**d**) σ_xy_; and (**e**) ε_max_ distributions.

**Figure 11 materials-17-02093-f011:**
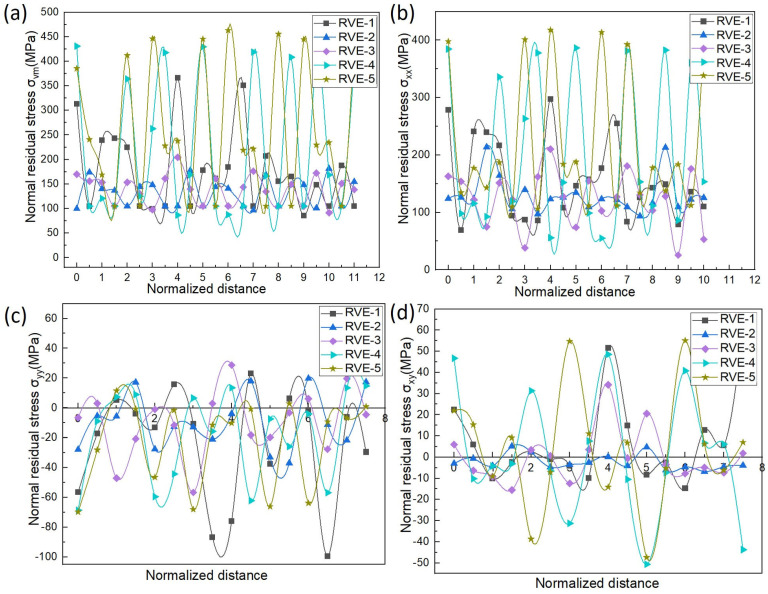
Variations in equivalent and normal residual stresses of RVE−1, RVE−2, RVE−3, RVE−4, and RVE−5: (**a**) σ_vm_; (**b**) σ_xx_; (**c**) σ_yy_; and (**d**) σ_xy._

**Figure 12 materials-17-02093-f012:**
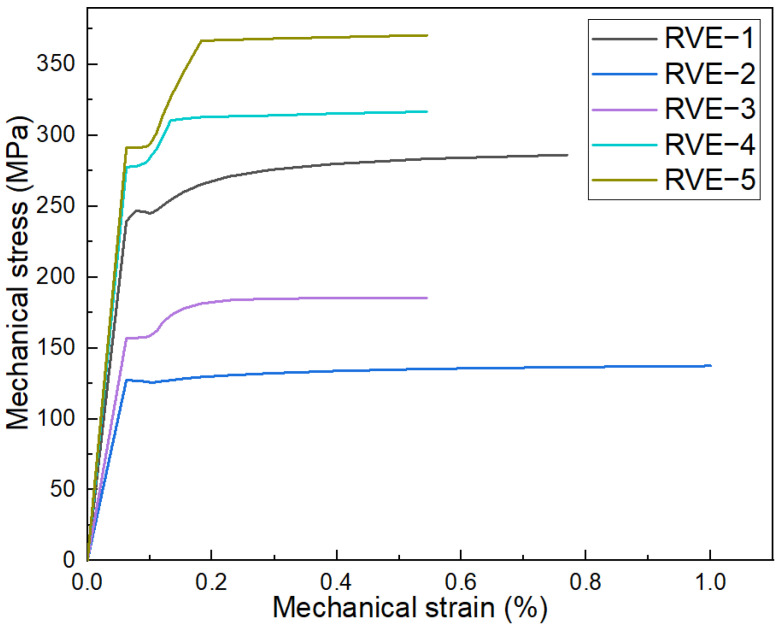
Stress–strain curves of RVE−1, RVE−2, RVE−3, RVE−4, and RVE−5 case models.

**Figure 13 materials-17-02093-f013:**
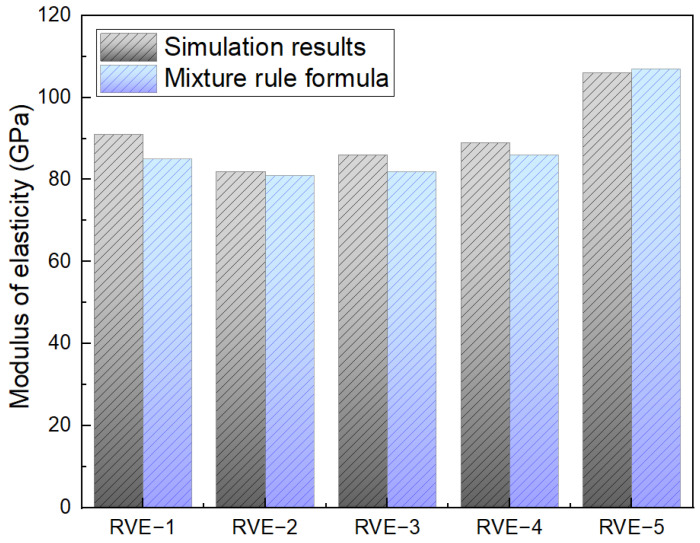
Young’s modulus of RVE−1−5 by simulation results and mixture rule formula.

**Figure 14 materials-17-02093-f014:**
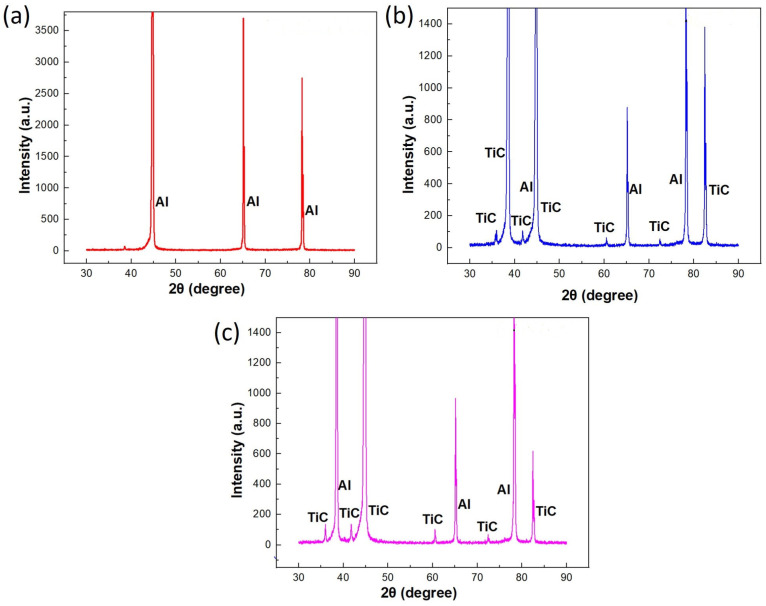
XRD patterns of Al–TiCp (**a**) initial material, (**b**) ARB-5, and (**c**) CR-5.

**Figure 15 materials-17-02093-f015:**
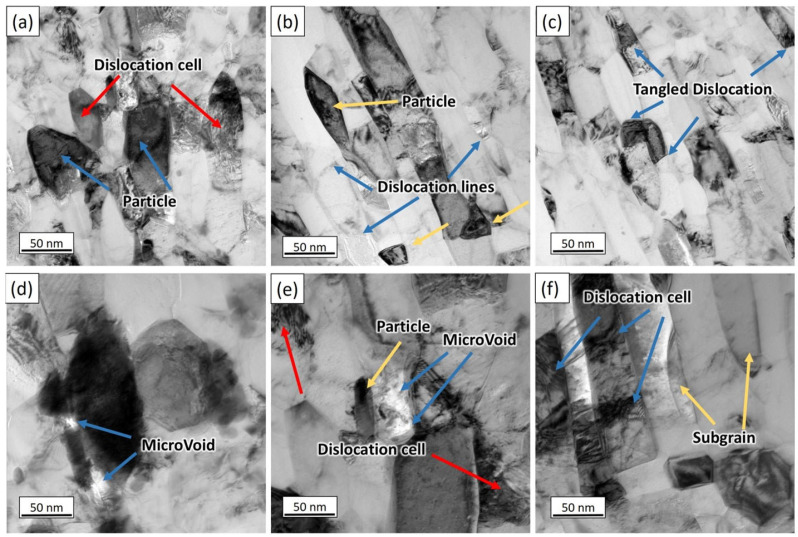
TEM images of AMCs after rolling reduction via ARB and CR processes: (**a**) dislocation cells and particles, (**b**) dislocation lines and particles, (**c**) tangled dislocation, (**d**) microvoids, (**e**) microvoids and particles, and (**f**) subgrain and dislocation cells after CR-5 sample.

**Figure 16 materials-17-02093-f016:**
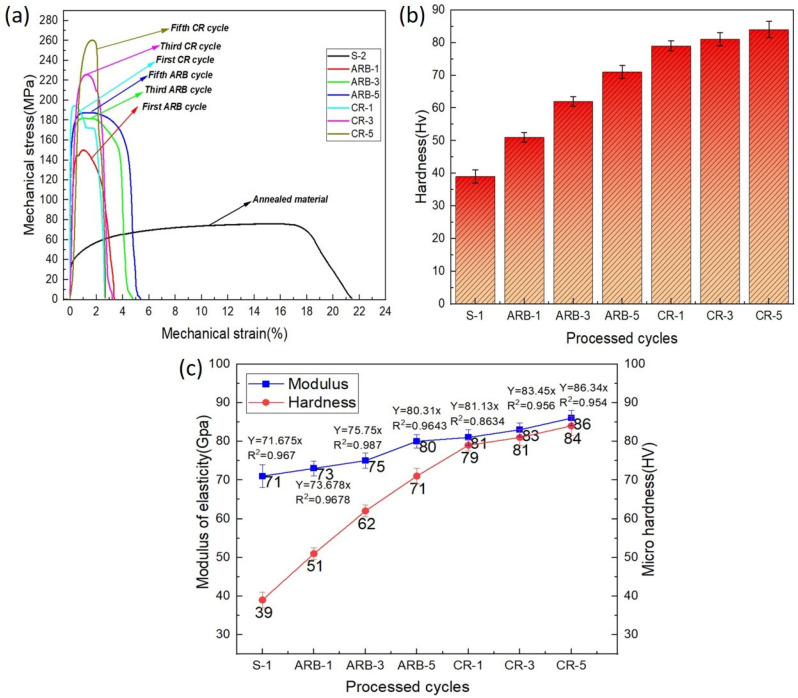
Mechanical properties of Al–TiCp: (**a**) stress–strain curves, (**b**) hardness, and (**c**) modulus of elasticity with hardness.

**Figure 17 materials-17-02093-f017:**
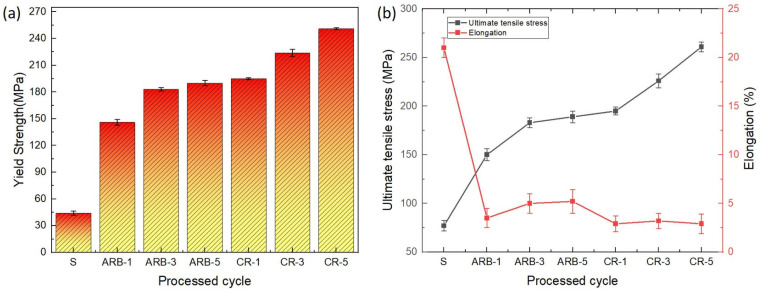
Al–TiC matrix: (**a**) yield stress and (**b**) tensile strength with elongation resulting from ARB in different cycles.

**Figure 18 materials-17-02093-f018:**
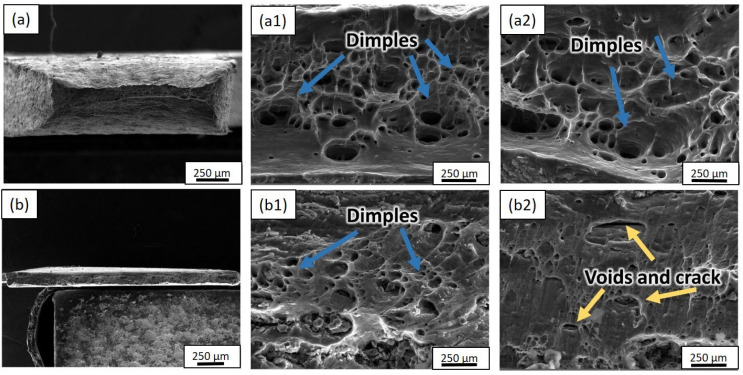
Diagram of SEM tensile fracture surfaces of samples: (**a**), (**a1**), and (**a2**) for initial aluminum material; (**b**), (**b1**), and (**b2**) for final composite after CR-5.

**Figure 19 materials-17-02093-f019:**
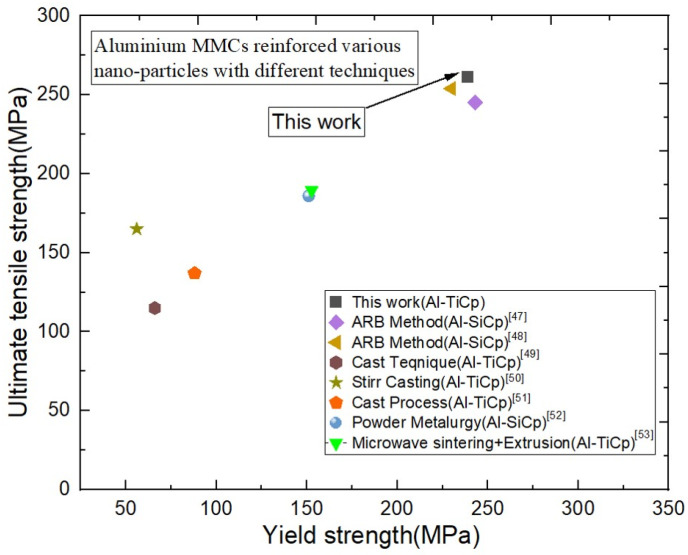
Comparison of various nanoparticle-reinforced MMCs between references [48,49,50,51,52,53,54] and current study.

**Table 1 materials-17-02093-t001:** List of abbreviations.

Abbreviations	Explanation
AMMCs	Aluminum Metal Matrix Composites
ARB	Accumulative Roll Bonding
AMCs	Aluminum Matrix Composites
CR	Cryorolling
FEA	Finite Element Analysis
RVE	Representative Volume Element
SEM	Scanning Electron Microscopy
TEM	Transmission Electron Microscopy
OM	Optical microscope
XRD	X-ray diffraction
TiC	Titanium Carbide
σvm	Von Mises Stress
σxx	Normal Stress in the x-direction
σyy	Normal Stress in the y-direction
σxy	Shear Stress
εmax	Maximum Strain
Vf	Volume fraction

**Table 2 materials-17-02093-t002:** Aluminum sheet composition.

Elements	Al	Fe	Si	Cu	Mg	Mn	Ni	Ti	Zn
Wt%	99.18	0.40	0.15	0.05	0.05	0.05	0.0014	0.05	0.07

**Table 3 materials-17-02093-t003:** Material parameters of TiC particles and Al matrix.

Materials	AA1050	TiC
Elastic modulus (GPa)	70	497
Yield strength (MPa)	105	20 × 10^3^
Poisson ratio	0.33	0.21
Density(kg/m^3^)	2700	4930
Thermal expansion coefficient (α)/°C	23.1 × 10^−6^	6.6 × 10^−6^
Thermal conductivity (k) W/(m·K)	200	22

**Table 4 materials-17-02093-t004:** The computational and experimental outcomes of models.

Materials	Experimental	Simulation (RVE−1 Case Model)	Error Percentage (%)
Elastic modulus (GPa)	86 ± 2	91	5.4
Yield strength (MPa)	260 ± 13	279	6.8
Tensile strength (MPa)	239 ± 15	246	2.8

## Data Availability

The data presented in this study are available on request from the corresponding author.

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
