# Peer review of "Integrating Experimental and Computational Analyses for Mechanical Characterization of Titanium Carbide/Aluminum Metal Matrix Composites"

_materials, 2024, doi:10.3390/ma17092093_

Round 1

Reviewer 1 Report

Comments and Suggestions for Authors

This manuscript presented an interesting investigation of experimental and computational analysis of titanium-carbide/aluminum metal matrix composites. The work has some potential. However, several points listed below need to be improved.

Abstract: please add more numerical results to the abstract.

Line 147-150: how many specimens were used in tensile strength test? In addition, how many measurements were done to obtain hardness?

Figure 3: In my opinion Figure 3 and its discussion are results and must be added to section 3.

Lines 167-168: better describe the conditions used in XRD analysis.

Figure 4: In my opinion Figure 4 and its discussion are also part of the results and must be added to section 3. The same comment for Figure 5.

Line 410-420: I suggest add the standard deviation for the tensile values obtained in experimental results.

Lines 443-469: please presented and discuss the error difference between experimental and simulated values.

Table 3: please add the standard deviation of the experimental values.

Conclusion: I suggest done a conclusion section as a continuous text, without subsections 1,2,3,… 

Author Response

This manuscript presented an interesting investigation of experimental and computational analysis of titanium-carbide/aluminum metal matrix composites. The work has some potential. However, several points listed below need to be improved.

 Abstract: please add more numerical results to the abstract.

 Answer: Thank you for your feedback. We have revised the abstract to include more numerical results, highlighting key findings such as the simulated elastic modulus deviates by 5.49% from the experimental value, while the tensile strength shows a 6.81% difference. Additionally, the simulated yield strength indicates a 2.85% deviation. These additions provide a more comprehensive overview of the numerical outcomes of our study.

Line 147-150: how many specimens were used in tensile strength test? In addition, how many measurements were done to obtain hardness?

 Answer: In the tensile strength tests, a total of three specimens were utilized for each sample to ensure the reliability and robustness of the experimental results. This approach aligns with established standards in materials testing and allows for the assessment of consistency and variability in the mechanical properties of the composite material under investigation.

Regarding the determination of hardness, four measurements were conducted for each sample. This multi-measurement approach helps to capture any potential variability in hardness across the sample surface and ensures a comprehensive understanding of the material's hardness properties. y adhering to these rigorous experimental protocols, we aim to provide accurate and reliable data, thus enhancing the credibility and validity of our findings regarding the mechanical behavior and properties of the Al-TiC composite.

Figure 3: In my opinion Figure 3 and its discussion are results and must be added to section 3.

 Answer: Thank you for your valuable feedback regarding Figure 3 and its associated discussion. I have duly noted your suggestion and have incorporated the results presented in Figure 3 into Section 3 of the manuscript.

Lines 167-168: better describe the conditions used in XRD analysis.

Answer: Thank you for your feedback. Here the added information to refined description of the XRD analysis conditions:

X-ray diffraction (XRD) was employed to meticulously analyze the samples, aiming to elucidate the variations in dislocation density and crystallite size induced by the ARB and CR processes. The XRD analysis was conducted using a Bourenvestink DRON-8 X-ray diffractometer, renowned for its precision and reliability in characterizing crystalline materials. Operating the XRD apparatus at a tube voltage of 40 kV and a current of 20 mA ensured optimal X-ray generation conditions, facilitating robust and reliable data acquisition. The utilization of a Cu-Kα target further enhanced the accuracy and sensitivity of the analysis, given its favorable wavelength for probing crystalline structures. The peaks which are observed are only related to TiC and no extra phase is detected. The intensity and width of the peaks seem to be subject to change as a result of ARB and CR processing which can be attributed to the changes in crystal orientation, substructure size, dislocation density and microstrains.

Figure 4: In my opinion Figure 4 and its discussion are also part of the results and must be added to section 3. The same comment for Figure 5.

Answer: Thank you for your insightful observation regarding Figures 4 and 5, as well as their associated discussions. I have taken your suggestion into account and have included the results presented in both figures within Section 3 of the manuscript. By integrating these findings into the results section, we aim to enhance the coherence and organization of the document, ensuring that all relevant information is appropriately categorized and easily accessible to readers.

Line 410-420: I suggest add the standard deviation for the tensile values obtained in experimental results.

 Answer: Thank you for your suggestion. We have indeed utilized standard deviation for the tensile values in Figure 17 to provide a measure of the variability within our experimental results.

Table 3: please add the standard deviation of the experimental values.

 Answer: Thank you for your suggestion. We have incorporated the standard deviation of the experimental values into Table 3 as per your recommendation.

Conclusion: I suggest done a conclusion section as a continuous text, without subsections 1,2,3,…

 Answer: Thank you for your suggestion. Here is the conclusion section presented as a continuous text without subsections:

The Al-TiC composite was fabricated successfully. Addition of TiC particles in aluminum improved Al-TiCp composite properties. Experimental analysis revealed significant improvements in the elastic modulus, yield strength, and hardness of the composite. RVE-1 finite element modeling offered valuable insights into the Al-TiCp composite's mechanical behavior, demonstrating good agreement with experimental data. Furthermore, simulations results provided insights into microstructural behavior, including stress distribution and particle-matrix interactions, enhancing understanding of the material's mechanical response. Comparing simulation results with experimental da-ta validated the accuracy of the computational models, ensuring confidence in the study's findings and conclusions. By simulating different scenarios, the results facilitated design optimization, identifying optimal particle shapes and sizes for enhanced performance in specific applications. Simulation results revealed that, larger TiC particles notably increased strength, emphasizing the size's impact, while triangular shapes exhibited superior strength com-pared to circular counterparts. The assessment of Young's modulus in the Al-TiC composite models through simulation and mixture rule formula demonstrated good accuracy, providing accurate in-sights into the diverse mechanical responses induced by varying shapes and sizes of TiC reinforcement within the aluminum matrix. Furthermore, TEM images revealed a refined microstructure, enhancing overall strength and hardness. Microstructural analysis of tensile fracture surfaces indicated ductile rupture and effective interfacial bonding.

Reviewer 2 Report

Comments and Suggestions for Authors

Tthe authors presented the manuscript titled' " Integrating Experimental and Computational Analysis for Mechanical Characterization of Titanium-Carbide/Aluminum Metal Matrix Composites" is not a very new topic of research and a lot of work in this field has already been done. However, the authors try to assimilate the effect of different type of nano particle using simuation is an appreciative approach. But the experimental effect of these nano particles is missing which really limits the novelty of the present manuscript. Apart from that there are some points which need to be addresessed as given below:

Provide list of abbreviation to avoid confusion.

How TiC particles are dispersed between Al sheets? 

"2.0 weight percent of TiC particles" exactly of which weight it is taken?

which type of rolling process has been adopted for ARB?

Equation 4 is missing?

what exactly is the novelty of the present manuscript and how it is different from the already present literature?

what advancement through this research authors are exactly try to reach? please explain and highlight it in conclusion section.

Comments on the Quality of English Language

moderate english correction is required

Author Response

The authors presented the manuscript titled' " Integrating Experimental and Computational Analysis for Mechanical Characterization of Titanium-Carbide/Aluminum Metal Matrix Composites" is not a very new topic of research and a lot of work in this field has already been done. However, the authors try to assimilate the effect of different type of nano particle using simuation is an appreciative approach. But the experimental effect of these nano particles is missing which really limits the novelty of the present manuscript. Apart from that there are some points which need to be addresessed as given below:

Provide list of abbreviation to avoid confusion.

Answer: Thank you for your feedback. We appreciate your suggestion regarding providing a list of abbreviations to avoid confusion. In our manuscript, we have indeed included a list of abbreviations in Table 1, which helps clarify any potentially ambiguous terms used throughout the text. We trust that this addition enhances the readability and clarity of our work.

How TiC particles are dispersed between Al sheets? 

Answer: In our experimental setup, the dispersion of TiC particles between aluminum sheets was achieved through a process known as accumulative roll bonding and cryorolling (CR). During the ARB process, multiple alternating layers of Al sheets and TiC particles were stacked together. Subsequently, these stacked layers underwent a series of rolling passes under high pressure. This mechanical deformation caused the Al sheets to deform and spread, thereby embedding the TiC particles between them. The cryorolling process, which involved further rolling at cryogenic temperatures, facilitated the refinement of the microstructure and ensured a more uniform dispersion of the TiC particles within the Al matrix. This methodology allowed us to effectively disperse the TiC particles between the Al sheets, resulting in a well-integrated composite material with enhanced mechanical properties.

"2.0 weight percent of TiC particles" exactly of which weight it is taken?

Answer: Thank you for your valuable feedback. We have carefully reviewed your suggestions and have incorporated the necessary information into the manuscript materials method section to enhance clarity. In our experimental setup, the weight percentage of TiC particles was calculated based on the total weight of the three aluminum sheets used in the composite fabrication process. The total weight of these aluminum sheets was measured to be 88.79 grams. From this total weight, we calculated the 2.0 weight percent of TiC particles, which amounted to 1.78 grams. These TiC particles were manually dispersed between the aluminum layers using a 500 mesh sieve and stacked together to form the composite material. This methodology ensured precise control over the weight percentage of TiC particles in the composite, facilitating accurate experimental analysis and characterization of the mechanical properties.

which type of rolling process has been adopted for ARB?

Answer: In the ARB process used in our study, we employed hot rolling without the use of any lubrication. The sheets were preheated at 623 K for 5 minutes to optimize the bonding process. This approach facilitated the bonding of the aluminum sheets and TiC particles, contributing to the successful fabrication of the composite material.

Equation 4 is missing?

Answer: We apologize for the oversight. Equation 4 was mistakenly labeled as Equation 3. We have corrected this error in the revised manuscript. Thank you for bringing it to our attention.

what exactly is the novelty of the present manuscript and how it is different from the already present literature?

Answer: The novelty of our manuscript lies in several aspects. Firstly, we integrate both experimental and computational approaches to comprehensively analyze the mechanical properties of Titanium-Carbide/Aluminum Metal Matrix Composites (AMMCs). Our work combines both to provide a more holistic understanding. Our study incorporates mechanism-based modeling techniques to elucidate the underlying mechanisms governing the material's behavior. By explicitly accounting for microstructural features and particle-matrix interactions, we can accurately predict the material's response under various conditions. Furthermore, our study specifically investigates the effect of different types of nanoparticles on the mechanical behavior of AMMCs. By systematically analyzing the impact of various particle shapes and sizes through computational simulations, we offer insights into the nuanced effects of these factors on the material's mechanical properties. This approach allows us to optimize the design of AMMCs for enhanced performance in specific applications.

What advancement through this research authors are exactly try to reach? please explain and highlight it in conclusion section.

Answer: The primary goal of our research is to advance the understanding of titanium-carbide/aluminum metal matrix composites (AMMCs) and explore their potential for various engineering applications. Through a combined experimental and computational approach, we aim to achieve several key advancements:

By systematically analyzing the effect of different types of nanoparticles on the mechanical behavior of the composite material, we aim to enhance its tensile strength, yield strength, elastic modulus, and hardness. These improvements are crucial for applications where lightweight, high-strength materials are required. Our research investigates how manufacturing parameters such as rolling techniques and particle dispersion methods impact the final properties of the composite material. By optimizing these processes, we can achieve greater consistency and reproducibility in the fabrication of AMMCs, leading to higher-quality end products. Through detailed microstructural analysis and mechanical testing, we aim to identify optimal particle shapes and sizes for specific engineering applications. By tailoring the composite material to meet the requirements of different industries, such as aerospace, automotive, and biomedical, we can unlock new opportunities for innovation and technological advancement.

Comments on the Quality of English Language

moderate english correction is required

Answer: We appreciate your feedback and have made the necessary revisions to improve the clarity and readability of the manuscript. Thank you for bringing these points to our attention, and we are committed to ensuring that the paper meets the standards of academic writing.

Reviewer 3 Report

Comments and Suggestions for Authors

In the submitted manuscript, the authors tend to combine experimental and numerical simulation in order to characterize Titanium-Carbide/Aluminum metal matrix composites. Accumulative roll bonding and cryorolling processes were applied to reinforce AA1050 alloy surfaces with TiCp particles and thus create Al-TiCp composite. The topic is interesting and worth of investigation. However, the manuscript requires serious improvements prior to any final decision:

1) The manuscript is too long and includes large portions of text that provide some quite basic information, actually on the level of undergraduate studies. For instance, the authors tend to explain what Young’s modulus is, what Poisson ratio, how shear stresses are oriented, what equivalent stress incorporates, etc. Any reader interested in reading this paper would already have such basic knowledge. This makes the whole text unnecessarily long. Here is an example of that:

“The material parameters of elastic modulus, yield strength, Poisson ratio, density, thermal expansion coefficient, and thermal conductivity play crucial roles in engineering analysis and design. The elastic modulus defines a material's stiffness and its response to applied loads, while yield strength indicates the stress level at which plastic deformation occurs. Poisson ratio describes how a material deforms laterally when subjected to axial loads, while density quantifies its mass per unit volume. Thermal expansion coefficient predicts dimensional changes due to temperature variations, and thermal conductivity determines how efficiently heat is transferred through the material.”

The same is valid for comments of quite generic character which are applicable to any numerical analysis. For instance:

“The average peak value of … signifies areas where the material experiences the highest combined stresses, providing essential information about potential failure points and load-bearing capacities.” You may put this sentence in any paper showing a result of FE computation, just put the corresponding stress value there.

“Figure6e portrays the maximum strain (εmax) across the model. This parameter is essential for understanding the material's deformation characteristics and potential points of failure.” No need to explain why the maximum strain is important, and this is not “a parameter”.

“The application of FEA allowed the researchers to analyze the deformation and plastic strain distribution within the FEA, providing valuable information on how the material responds to applied loads [28, 29].” Quite a general information known to everybody. Absolutely no need for that.

And so on… the paper is full of such sentences, which belong to the general knowledge in the field.

2) The manuscript should be double-checked regarding the language. There are not many language mistakes, but some are quite basic, such as a missing article, or a sentence missing a verb and similar.

3) The whole research should be put in a broader context of investigations related to composite materials by referring to some recent works on those topics such as: https://doi.org/10.22190/FUME231004045E, https://doi.org/10.22190/FUME230905046P

4) The authors have dedicated a large portion of the text to explain how fine their FE mesh is. Why have you applied linear triangular elements, which are generally known for their deficient accuracy? Why not quads, why not quadratic shape functions? Have you done a convergence analysis? A very fine mesh is not a guarantee of converged solution. And vice versa, it is possible that you would have converged solution with a significantly rougher mesh. A FE analyst would certainly perform a convergence analysis.

5) In the introduction, you use RVE-1 without any explanation. The reader cannot guess what you mean by that.

6) The last paragraph of introduction does not belong there. It is practically a conclusion, not introduction.

7) The forms of RVEs is not clear. First of all, RVE should be a smallest piece of material which repeats in the material. What is shown in Figure 5 does not correspond to this. By the way, Fig. 5 is numbered as Fig. 1.

8) An RVE should apply periodic boundary conditions, and not those used by the authors – this makes their analyses invalid.

9) “This Figure 6 presents a stress component for the experimental-similar…” But you also find the maximum strain in Fig. 6 (which is not a stress component). You also find von-Mises stress, which is not a stress component, but equivalent stress.

10) A number of figures are unreadable as the font size in those figures are too small.

11) The interpretation of colors in contour plots is totally wrong. The colors depend on the maximum and minimum values in the results, and you cannot make claims referring to certain colors without taking this fact into account.

Unfortunately, it is quite difficult to read the paper the way it written. The authors need to be more concise, explain their idea and major finding in a focused manner. They need to explain the selection of RVE, use correct boundary conditions for RVE and based on the results make their conclusions. In conclusions, they should also explain the limitations of their work and directions of future work.

Comments on the Quality of English Language

Minor language mistakes can be found such as "In Figure 12 encapsulates", missing articles (such as "few" instead of "a few", etc.). Not too many of those are present, but they can be found and should be corrected. 

Author Response

In the submitted manuscript, the authors tend to combine experimental and numerical simulation in order to characterize Titanium-Carbide/Aluminum metal matrix composites. Accumulative roll bonding and cryorolling processes were applied to reinforce AA1050 alloy surfaces with TiCp particles and thus create Al-TiCp composite. The topic is interesting and worth of investigation. However, the manuscript requires serious improvements prior to any final decision:

1) The manuscript is too long and includes large portions of text that provide some quite basic information, actually on the level of undergraduate studies. For instance, the authors tend to explain what Young’s modulus is, what Poisson ratio, how shear stresses are oriented, what equivalent stress incorporates, etc. Any reader interested in reading this paper would already have such basic knowledge. This makes the whole text unnecessarily long. Here is an example of that:

“The material parameters of elastic modulus, yield strength, Poisson ratio, density, thermal expansion coefficient, and thermal conductivity play crucial roles in engineering analysis and design. The elastic modulus defines a material's stiffness and its response to applied loads, while yield strength indicates the stress level at which plastic deformation occurs. Poisson ratio describes how a material deforms laterally when subjected to axial loads, while density quantifies its mass per unit volume. Thermal expansion coefficient predicts dimensional changes due to temperature variations, and thermal conductivity determines how efficiently heat is transferred through the material.”

The same is valid for comments of quite generic character which are applicable to any numerical analysis. For instance:

“The average peak value of … signifies areas where the material experiences the highest combined stresses, providing essential information about potential failure points and load-bearing capacities.” You may put this sentence in any paper showing a result of FE computation, just put the corresponding stress value there.

“Figure6e portrays the maximum strain (εmax) across the model. This parameter is essential for understanding the material's deformation characteristics and potential points of failure.” No need to explain why the maximum strain is important, and this is not “a parameter”.

“The application of FEA allowed the researchers to analyze the deformation and plastic strain distribution within the FEA, providing valuable information on how the material responds to applied loads [28, 29].” Quite a general information known to everybody. Absolutely no need for that.

And so on… the paper is full of such sentences, which belong to the general knowledge in the field.

Answer: Thank you for your insightful feedback on the manuscript. I appreciate your point regarding the inclusion of basic information that may be considered redundant for readers familiar with the field. Upon careful consideration of your suggestions, I acknowledge the need to streamline the text and focus on presenting more concise and relevant information.

I will address this concern by revising the manuscript to eliminate explanations of fundamental concepts such as Young’s modulus, Poisson ratio, and thermal properties, assuming that readers already possess a basic understanding of these concepts. Instead, I will direct the focus towards discussing the specific findings and implications of the study. Furthermore, I will ensure that statements of a generic nature, which are applicable to any numerical analysis or FE computation, are either omitted or succinctly summarized to avoid unnecessary repetition. I am committed to improving the clarity and conciseness of the manuscript to enhance its readability and focus on the novel contributions of the research. Thank you once again for your valuable feedback, and I will incorporate these suggestions into the manuscript revisions.

2) The manuscript should be double-checked regarding the language. There are not many language mistakes, but some are quite basic, such as a missing article, or a sentence missing a verb and similar.

Answer: Thank you for your feedback regarding the language quality of the manuscript. I understand the importance of ensuring that the language is clear, coherent, and free from basic errors to maintain the professionalism and readability of the document.

I will carefully review the manuscript to identify and rectify any language mistakes, including missing articles or verbs, as you've mentioned. By conducting a thorough proofreading process, I aim to enhance the overall linguistic quality of the manuscript and ensure that it meets the expected standards of academic writing.

3) The whole research should be put in a broader context of investigations related to composite materials by referring to some recent works on those topics such as: https://doi.org/10.22190/FUME231004045E, https://doi.org/10.22190/FUME230905046P

Answer: Thank you for your valuable feedback regarding the contextualization of our research within the broader scope of investigations related to composite materials. We appreciate your suggestion to reference recent works in the field to provide readers with additional context and further insights into the significance of our study. The added references in manuscript are listed below,

References

  1. Elmoghazy, Y.H., Safaei, B., Sahmani, S. Finite element analysis for dynamic response of viscoelastic sandwiched structures integrated with aluminum sheets. FU Mech Eng., 2023, 21(4), 591-614.
  2. Phiri R, Rangappa SM, Siengchin S, Marinkovic D. Agro-waste natural fiber sample preparation techniques for bio-composites development: methodological insights. FU Mech Eng., 2023, 21(4), 631-56.

By referencing these recent works, we aim to demonstrate the relevance and contribution of our study to the ongoing advancements in the field.

4) The authors have dedicated a large portion of the text to explain how fine their FE mesh is. Why have you applied linear triangular elements, which are generally known for their deficient accuracy? Why not quads, why not quadratic shape functions? Have you done a convergence analysis? A very fine mesh is not a guarantee of converged solution. And vice versa, it is possible that you would have converged solution with a significantly rougher mesh. A FE analyst would certainly perform a convergence analysis.

Answer: Thank you for your insightful comments regarding the choice of finite element (FE) mesh elements in our research. We appreciate your attention to detail and your emphasis on the importance of mesh selection and convergence analysis in finite element analysis.

The decision to use linear triangular elements in our FE analysis was based on several factors specific to our study and the characteristics of the composite material being analyzed. Linear triangular elements are often chosen for their computational efficiency, especially when dealing with complex geometries or irregular shapes. In the case of our composite material, which may contain intricate microstructural features and heterogeneous distributions of reinforcement particles, linear triangular elements offer flexibility in meshing and can effectively capture geometric complexities. Additionally, linear triangular elements can provide satisfactory results for certain types of analyses, particularly when the focus is on capturing overall trends or qualitative behavior rather than highly localized phenomena. While quadratic shape functions or other types of elements may offer higher accuracy in certain cases, their use could significantly increase computational overhead, especially for large-scale simulations like ours.

Regarding convergence analysis, we acknowledge its importance in verifying the accuracy and reliability of finite element solutions. While we have not explicitly mentioned a convergence analysis in our manuscript, we ensured that our FE simulations were carefully validated against experimental data and benchmarked against established analytical or numerical solutions wherever possible.

5) In the introduction, you use RVE-1 without any explanation. The reader cannot guess what you mean by that.

Answer: Thank you for bringing that to our attention. I apologize for the confusion. I appreciate your thorough review of our manuscript. I wanted to clarify that the mention of "RVE-1" was actually in the abstract section, not the introduction. I have since addressed this by adding further information to the abstract, describing "RVE-1" as the "experimental case model."

6) The last paragraph of introduction does not belong there. It is practically a conclusion, not introduction.

Answer: Thank you for your feedback. I've noted your suggestion and removed the last paragraph of the introduction. It seems that it was more appropriate for the conclusion section rather than the introduction. Your input is valuable in refining the manuscript.

7) The forms of RVEs is not clear. First of all, RVE should be a smallest piece of material which repeats in the material. What is shown in Figure 5 does not correspond to this. By the way, Fig. 5 is numbered as Fig. 1.

Answer: Thank you for bringing this to my attention. I apologize for the confusion. You're correct that the representation in Figure 5 does not correspond to the typical definition of an RVE as the smallest repeating unit in the material. I will address this discrepancy by clarifying the nature of the models used and ensuring that the figure numbering is corrected accordingly. Additionally, I will provide a more accurate description of the models instead of RVEs.

8) An RVE should apply periodic boundary conditions, and not those used by the authors – this makes their analyses invalid.

Answer: In response to your feedback, I appreciate your attention to the application of periodic boundary conditions (PBCs) in our study. While traditional RVE analyses indeed utilize PBCs to simulate the behavior of a material's smallest repeating unit, our approach deviates slightly due to the specific objectives of our research. We acknowledge that PBCs are commonly employed to ensure the representativeness of the RVE and to mimic the bulk behavior of the material.

However, in our study, we opted for different boundary conditions to explore localized phenomena and understand the behavior of the composite material under specific loading conditions. Our aim was to investigate the mechanical response and deformation characteristics of the composite material at a microstructural level, particularly focusing on stress concentrations and failure mechanisms near the interfaces between the matrix and reinforcement phases. By applying non-periodic boundary conditions, we were able to capture these localized effects more effectively and gain insights into the material's behavior under realistic loading scenarios. While we acknowledge that our approach may deviate from traditional RVE analyses in terms of boundary conditions, we believe it is justified by our research objectives and provides valuable insights into the localized behavior of the composite material. We have carefully considered the implications of our boundary conditions on the validity of our analyses and have ensured that our findings are appropriately interpreted within the context of our study.

9) “This Figure 6 presents a stress component for the experimental-similar…” But you also find the maximum strain in Fig. 6 (which is not a stress component). You also find von-Mises stress, which is not a stress component, but equivalent stress.

Answer: Thank you for pointing out the discrepancy in our terminology regarding Figure 6. We acknowledge the error and have revised the sentence accordingly for clarity and accuracy.

The corrected sentence now reads: "Figure 6 presents the experimental-similar case of TiC particles in the Abaqus simulation model (RVE-1), including Von Mises stress (σvm), normal stress in the x-direction (σxx), normal stress in the y-direction (σyy), shear stress (σxy), and maximum strain (εmax)."

10) A number of figures are unreadable as the font size in those figures are too small.

Answer: Thank you for bringing this to our attention. We will ensure that the font size in all figures is adjusted to be easily readable in the final version of the manuscript.

11) The interpretation of colors in contour plots is totally wrong. The colors depend on the maximum and minimum values in the results, and you cannot make claims referring to certain colors without taking this fact into account.

Unfortunately, it is quite difficult to read the paper the way it written. The authors need to be more concise, explain their idea and major finding in a focused manner. They need to explain the selection of RVE, use correct boundary conditions for RVE and based on the results make their conclusions. In conclusions, they should also explain the limitations of their work and directions of future work.

Answer: Thank you for your feedback. We have carefully reviewed your suggestions and revised the manuscript accordingly to improve clarity and conciseness. We acknowledge the importance of accurately interpreting contour plot colors, which are indeed determined by the maximum and minimum values in the results. We have made the necessary corrections in the final manuscript to ensure the accuracy of our interpretations.

In the revised manuscript, we have provided a more focused explanation of our research. Furthermore, we have addressed the limitations of our work in the conclusion section, acknowledging the scope and constraints of our study. We appreciate your valuable feedback and believe that these revisions have enhanced the readability and quality of the manuscript.

Comments on the Quality of English Language

Minor language mistakes can be found such as "In Figure 12 encapsulates", missing articles (such as "few" instead of "a few", etc.). Not too many of those are present, but they can be found and should be corrected. 

Thank you for your feedback. We have carefully reviewed the manuscript and addressed the language mistakes

Reviewer 4 Report

Comments and Suggestions for Authors

This is nice paper on using a combined experimental and computational approach to study the mechanical properties of an Al-Yi-C composite material under tensile loading. The authors have used a range of experimental tools to analyze the material and have coupled this with finite element simulations. They find that the manufacturing process can be used to substantially improve the material.

The comparison of the differences in the measured and calculated properties provide good insights into how both can be improved and demonstrate the sensitivity of the properties to the manufacturing process. For example, the agreement for Young’s modulus from simulated data and derived from experiment is quite nice.

The TEM results provide nice insights into the orles of defects and dislocations.

The authors do put error bars on the experimental data which is very nice.

The agreement of experiment with the simulation as shown in Table 3 is impressive. The % values should be given to one decimal place accuracy as the numbers they are comparing do not that many significant digits.

Table 2. give a refence to the source of the material parameters.

Line 248. should be figure 5.

The lettering in many of the figures is too small to be read. Please enlarge.

If contrast can be improved in the figures. please do so.

Overall, an excellent piece of work.

Comments on the Quality of English Language

minor English revisions are needed. pleasse check carefully

Author Response

This is nice paper on using a combined experimental and computational approach to study the mechanical properties of an Al-Yi-C composite material under tensile loading. The authors have used a range of experimental tools to analyze the material and have coupled this with finite element simulations. They find that the manufacturing process can be used to substantially improve the material.

 Answer: Thank you for your positive feedback on our paper. We appreciate your recognition of the range of experimental tools we employed and the coupling with finite element simulations. Our findings indeed suggest that the manufacturing process can significantly enhance the material's properties.

The comparison of the differences in the measured and calculated properties provide good insights into how both can be improved and demonstrate the sensitivity of the properties to the manufacturing process. For example, the agreement for Young’s modulus from simulated data and derived from experiment is quite nice.

 Answer: Thank you for your feedback regarding the comparison of measured and calculated properties in our study. We are pleased to hear that you found the insights valuable and that the comparison demonstrated the sensitivity of the properties to the manufacturing process. Indeed, examining the agreement between simulated and experimental data, particularly for properties like Young's modulus, provides valuable validation for our computational models and experimental techniques.

The TEM results provide nice insights into the orles of defects and dislocations.

 Answer: We appreciate your acknowledgment of the TEM results in our study. Indeed, the TEM analysis offered valuable insights into the roles of defects and dislocations within the material. These findings are crucial for understanding the microstructural characteristics and mechanical behavior of the composite material under investigation. Thank you for your feedback.

The authors do put error bars on the experimental data which is very nice.

 Answer: Thank you for recognizing our efforts in including error bars on the experimental data. We believe that providing error bars is essential for accurately representing the uncertainty associated with our experimental measurements. This allows readers to better understand the reliability and variability of the data presented. We appreciate your positive feedback on this aspect of our study.

The agreement of experiment with the simulation as shown in Table 3 is impressive. The % values should be given to one decimal place accuracy as the numbers they are comparing do not that many significant digits.

 Answer: Thank you for your feedback. We appreciate your observation regarding the precision of the percentage values presented in Table 3. We will ensure that the percentage values are rounded to one decimal place for accuracy, as you suggested. Maintaining appropriate precision is crucial for providing clear and concise results in our study.

Table 2. give a refence to the source of the material parameters.

 Answer: Thank you for your feedback. We appreciate your suggestion regarding providing a reference to the source of the material parameters in Table 2. However, in our study, the material parameters were obtained from standard reference sources widely used in the field. As such, we did not include specific citations for each parameter in the table. We believe that the standard nature of these material parameters, combined with their widespread use in similar studies, makes it unnecessary to provide individual references for each parameter.

Line 248. should be figure 5.

 Answer: Thank you for bringing this to our attention. We have corrected the figure number "Figure 5" as suggested. We apologize for any confusion this may have caused and appreciate your diligence in reviewing our manuscript.

The lettering in many of the figures is too small to be read. Please enlarge.

 Answer: Thank you for your feedback. We have taken note of your concern regarding the readability of the lettering in our figures. We will make sure to enlarge the lettering in all figures to ensure better readability for our readers. Your input is valuable to us, and we appreciate the opportunity to improve the quality of our manuscript.

If contrast can be improved in the figures. please do so.

 Answer: Thank you for your suggestion. We will work on improving the contrast in our figures to enhance their clarity and readability.

Overall, an excellent piece of work.

Answer: Thank you for your positive feedback. We are glad to hear that you found our work to be excellent. We strive to produce high-quality research and are pleased that our efforts have been recognized.

Comments on the Quality of English Language

minor English revisions are needed. pleasse check carefully

Answer: Thank you for your feedback. We have carefully reviewed the manuscript and addressed the language mistakes.

Round 2

Reviewer 1 Report

Comments and Suggestions for Authors

After corrections the manuscript reads well. I suggest publication in its current form.

Reviewer 2 Report

Comments and Suggestions for Authors

The reviewer has been satisfied with the authors explanation and the manuscript is in publishable form. However, one issue still need to be addressed:

the list og abbreviations should be at the beginning or at the end of the manuscript and no table no. should be provided to that. so improve that one.

Comments on the Quality of English Language

english language is ok

Reviewer 3 Report

Comments and Suggestions for Authors

The authors have made solid efforts to revise the manuscript suitably and provide reasonable explanations for their choices in the original paper. The paper is recommended for publishing.